

# In silico candidate variant and gene identification using inbred mouse strains

Matthias Munz[1,*], Mohammad Khodaygani[1,*], Zouhair Aherrahrou[2], Hauke Busch[1] and Inken Wohlers[1]

[1] Medical Systems Biology Division, Lübeck Institute of Experimental Dermatology and Institute for Cardiogenetics, University of Lübeck, Lübeck, Germany
[2] Institute for Cardiogenetics, University of Lübeck, Lübeck, Germany
[*] These authors contributed equally to this work.

Corresponding authors
Hauke Busch,
Hauke.Busch@uni-luebeck.de
Inken Wohlers,
Inken.Wohlers@uni-luebeck.de

## ABSTRACT

Mice are the most widely used animal model to study genotype to phenotype relationships. Inbred mice are genetically identical, which eliminates genetic heterogeneity and makes them particularly useful for genetic studies. Many different strains have been bred over decades and a vast amount of phenotypic data has been generated. In addition, recently whole genome sequencing-based genome-wide genotype data for many widely used inbred strains has been released. Here, we present an approach for in silico fine-mapping that uses genotypic data of 37 inbred mouse strains together with phenotypic data provided by the user to propose candidate variants and genes for the phenotype under study. Public genome-wide genotype data covering more than 74 million variant sites is queried efficiently in real-time to provide those variants that are compatible with the observed phenotype differences between strains. Variants can be filtered by molecular consequences and by corresponding molecular impact. Candidate gene lists can be generated from variant lists on the fly. Fine-mapping together with annotation or filtering of results is provided in a Bioconductor package called MouseFM. In order to characterize candidate variant lists under various settings, MouseFM was applied to two expression data sets across 20 inbred mouse strains, one from neutrophils and one from CD4$^+$ T cells. Fine-mapping was assessed for about 10,000 genes, respectively, and identified candidate variants and haplotypes for many expression quantitative trait loci (eQTLs) reported previously based on these data. For albinism, MouseFM reports only one variant allele of moderate or high molecular impact that only albino mice share: a missense variant in the *Tyr* gene, reported previously to be causal for this phenotype. Performing in silico fine-mapping for interfrontal bone formation in mice using four strains with and five strains without interfrontal bone results in 12 genes. Of these, three are related to skull shaping abnormality. Finally performing fine-mapping for dystrophic cardiac calcification by comparing 9 strains showing the phenotype with eight strains lacking it, we identify only one moderate impact variant in the known causal gene *Abcc6*. In summary, this illustrates the benefit of using MouseFM for candidate variant and gene identification.

## INTRODUCTION

Mice are the most widely used animal models in research. Several factors such as small size, low cost of maintain, and fast reproduction as well as sharing disease phenotypes and physiological similarities with human makes them one of the most favourable animal models (*Uhl & Warner, 2015*). Inbred mouse strains are strains with all mice being genetically identical, i.e., clones, as a result of sibling mating for many generations, which results in eventually identical chromosome copies. When assessing genetic variance between mouse strains, the genome of the most commonly used inbred strain, called Black 6J (C57BL/6J) is typically used as reference and variants called with respect to the Black 6J mouse genome. For inbred mouse strains, variants are homozygous by design.

*Grupe et al. (2001)* published impressive results utilizing first genome-wide genetic data for in silico fine-mapping of complex traits, "reducing the time required for analysis of such [inbred mouse] models from many months down to milliseconds". Darvasi commented on this paper that in his opinion, the benefit of in silico fine-mapping lies in the analysis of monogenic traits and in informing researchers prior to initiating traditional breeding-based studies. In 2007, with *Cervino et al. (2007)* he suggested to combine in silico mapping with expression information for gene prioritization using 20,000 and 240,000 common variants, respectively. Since then, the general approach has been applied successfully and uncovered a number of genotype-phenotype relationships in inbred mice (*Liao et al., 2004*; *Zheng, Dill & Peltz, 2012*; *Hall & Lammert, 2017*; *Mulligan et al., 2019*). Nonetheless, to the best of our knowledge, there is to date no tool publicly available that implements the idea and which allows to analyze any phenotype of interest. Such a tool is particularly helpful now that all genetic variation between all commonly used inbred strains is known at base pair resolution (*Doran et al., 2016*; *Keane et al., 2011*).

At the same time, in the last years huge amounts of mouse phenotype data were generated, often in collaborative efforts and systematically for many mouse strains. Examples are phenotyping undertaken by the International Mouse Phenotyping Consortium (IMPC) (*Dickinson et al., 2016*; *Meehan et al., 2017*) or lately also the phenotyping of the expanded BXD family of mice (*Ashbrook et al., 2021*). Data are publicly available in resources such as the mouse phenome database (MPD) (*Bogue et al., 2018*) (https://www.mousephenotype.org) or the IMPC's website (*Dickinson et al., 2016*) (https://phenome.jax.org). Other websites such as Mouse Genome Informatics (MGI) (http://www.informatics.jax.org) or GeneNetwork (*Mulligan et al., 2017*) (https://www.genenetwork.org) also house phenotype data together with web browser-based functionality to investigate genotype-phenotype relationships.

Several of the aforementioned resources allow the user to interactively query genotypes for 70 user-selected inbred mouse strains for input genes or genetic regions. Moreover, the variant browser in GeneNetwork allows also comparison of genotypes between strains, however, data can only be extracted gene- or region-wise and is not accessible programmatically. No current resource thus provides the functionality to extract genome-wide all variants that are different between two user-specified groups of inbred mouse strains. Such information can be used for in silico fine-mapping and for the identification
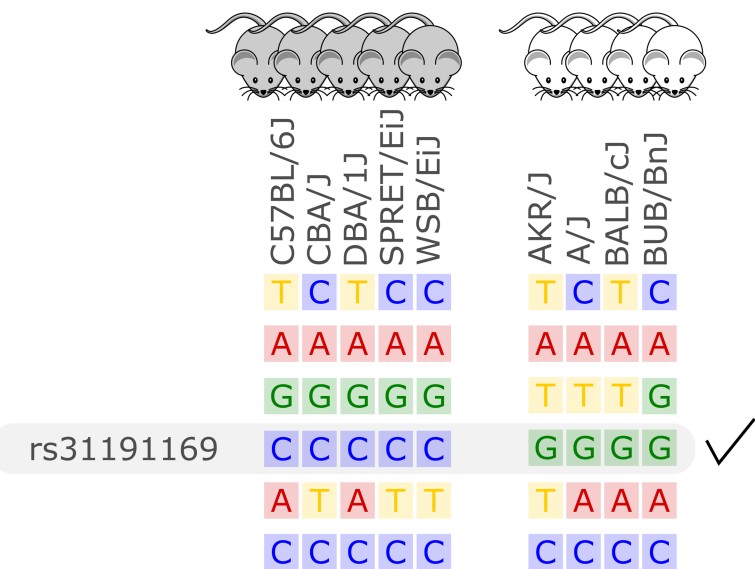

**Figure 1 Illustration of the in silico fine-mapping approach.** Every row represents a variant site and every column one inbred mouse strain. In this example, the phenotype is albinism and four strains are albinos and five are not. Displayed are six variants, but only one variant, rs31191169, has consistently different alleles between the albino and the other mice (G allele is here linked to albinism). With option thr2=1 in the MouseFM package, one discordant strain would be allowed in the second strain group and the variant in the row above rs31191169 would also be returned.

of candidate genes and variants underlying a phenotypic trait. Further, such a catalog of genetic differences between groups of strains is very useful prior to designing mouse breeding-based experiments e.g., for the identification or fine-mapping of quantitative trait loci (QTL).

## METHODS

### Fine-mapping approach

Unlike previous approaches for in silico fine-mapping, here we are using whole genome sequencing-based variant data and thus information on all single nucleotide variation present between inbred strains. Due to the completeness of this variant data, we do not need to perform any statistical aggregation of variant data over genetic loci, but simply report all variant sites with different alleles between two groups of inbred strains. That is, we report all variant sites with alleles compatible with the observed phenotype difference, see Fig. 1 for an illustration.

In the case of a binary phenotype caused by a single variant, this causal variant is one of the variants that has a different allele in those strains showing the phenotype compared to those strains lacking the phenotype. This is the case for example for albinism and its underlying causal variant rs31191169, used in Fig. 1 for illustration and discussed later in detail.

This in silico fine-mapping approach can reduce the number of variants to a much smaller set of variants that are compatible with a phenotype. The more inbred strains are
phenotyped and used for comparison, the more variants can be discarded because they are not compatible with the observed phenotypic difference.

In the case of a quantitative phenotype, the fine-mapping can be performed in two ways. The first option is to obtain genetic differences between strains showing the most extreme phenotypes. The second option is binarization of the phenotype by applying a cutoff. Since in these cases allele differences of variants affecting the trait may not be fully compatible with an artificially binarized phenotype, fine-mapping is provided with an option that allows alleles of a certain number of strains to be incompatible with the phenotype, see Fig. 1 for an example.

Two important, related aspects need to be considered with respect to the in silico fine-mapping approach implemented in MouseFM: (i) power and (ii) significance of the MouseFM candidates with respect to chance findings. With respect to (i): The suggested fine-mapping approach considerably gains power when increasing the number of inbred strains with phenotype data available. This is the result of an explosion of the number of possible genotype combinations across the analyzed strains. Figure 2 shows the number of possible genotype combinations. If, e.g., for a Mendelian trait, only one combination is compatible with the phenotype, it is increasingly unlikely to observe this combination by chance when the number of strains increases. Based on these theoretical considerations, we recommend using MouseFM for more than eight phenotyped strains. The number of actual genotype combinations for a given set of inbred strains is less than the maximum depicted in Fig. 2, because of kinship between strains. One favourable extreme are two phenotypic groups of overall closely related strains: only few variants differ between the groups and will be returned by MouseFM. The opposite extreme are groups of inbred strains closely related only within their phenotypic group, but not across groups: many variants will differ and be returned by MouseFM. With respect to (ii): For a low number of strains, a random split may result in a similar number of candidate variants compared to a split by phenotype and false-positive candidates increase. The important property is though, that in a split by phenotype, true positives will be among the candidates and once the number of phenotyped strains increases, the candidate set becomes smaller while still including true positives.

## Variant data

The database used by this tool was created based on the genetic variants database of the Mouse Genomes Project (https://www.sanger.ac.uk/science/data/mouse-genomes-project) of the Wellcome Sanger Institute. It includes whole genome sequencing-based single nucleotide variants of 36 inbred mouse strains which have been compiled by *Keane et al. (2011)*, see ftp://ftp-mouse.sanger.ac.uk/current_snps/mgp.v5.merged.snps_all.dbSNP142.vcf.gz. for the accession code and sources. This well designed set of inbred mouse strains for which genome-wide variant data is available includes classical laboratory strains (C3H/HeJ, CBA/J, A/J, AKR/J, DBA/2J, LP/J, BALB/cJ, NZO/HlLtJ, NOD/ShiLtJ), strains extensively used in knockout experiments (129S5SvEvBrd, 129P2/OlaHsd, 129S1/SvImJ, C57BL/6NJ), strains used commonly for a range of diseases (BUB/BnJ, C57BL/10J, C57BR/cdJ, C58/J, DBA/1J, I/LnJ, KK/HiJ, NZB/B1NJ, NZW/LacJ, RF/J, SEA/GnJ,

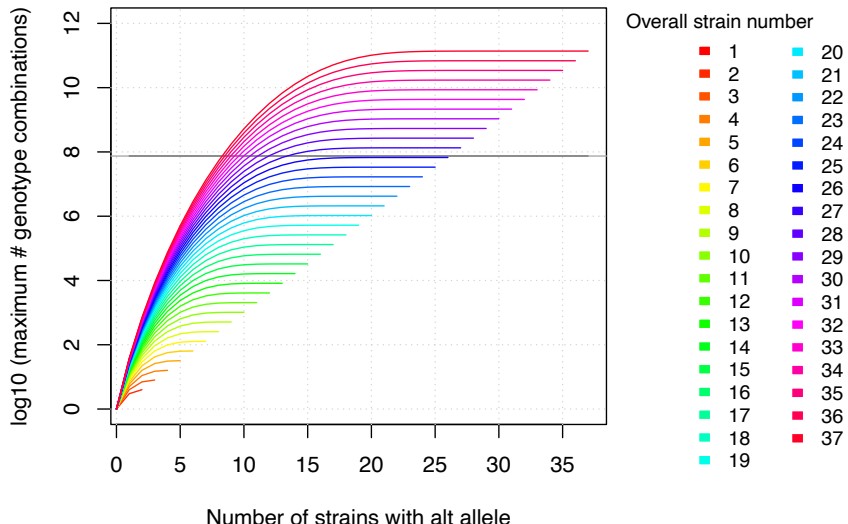

**Figure 2** **The maximum number of genotype combinations for an overall number of inbred strains $n$ including up to $k$ alternative alleles is given by $\sum_{k=1}^{n}\sum_{j=1}^{k}\binom{k}{j}$ and grows exponentially with respect to the overall number of inbred strains.** Further the more evenly the alleles are divided among these overall strains, the larger the corresponding number of genotype combinations. The gray horizontal line denotes the number of variants in MouseFM ($n = 74,480,058$). For more than 26 strains, the maximum number of genotype combinations are larger than the number of variant positions, and it is thus extremely unlikely to observe a phenotype-compatible combination by chance. For 10 and more strains, there is a maximum of more than 1000 genotype combinations, which reduces the probability of a phenotype-compatible combination already considerably. The number of actual, observed genotype combinations depends on the particular inbred strains used and, importantly, on their kinship.

ST/bJ) as well as wild-derived inbred strains from different mouse taxa (CAST/EiJ, PWK/PhJ, WSB/EiJ, SPRET/EiJ, MOLF/EiJ). Genome sequencing, variant identification an characterization of 17 strains was performed by *Keane et al. (2011)* and of 13 strains by *Doran et al. (2016)*. We downloaded the single nucleotide polymorphism (SNP) VCF file (https://www.sanger.ac.uk/data/mouse-genomes-project). Overall, it contains 78,767,736 SNPs, of which 74,873,854 are autosomal. The chromosomal positions map to the mouse reference genome assembly GRCm38 which is based on the Black 6J inbred mouse strain and by definition has no variant positions.

Low confidence, heterozygous, missing and multiallelic variants vary by strain, in sum they are typically less than 5% of the autosomal variants (Fig. 3, Table S1). Exceptions are for example the wild-derived inbred strains, for which variant genotypes excluded from the database reach a maximum of 11.5% for SPRET/EiJ. There are four strains that are markedly genetically different from each other and all remaining strains, these are the wild-derived, inbred strains CAST/EiJ, PWK/PhJ, SPRET/EiJ and MOLF/EiJ, see Fig. 3A. These four strains also show the highest number of missing and multiallelic genotypes (Fig. 3B and Table S1).

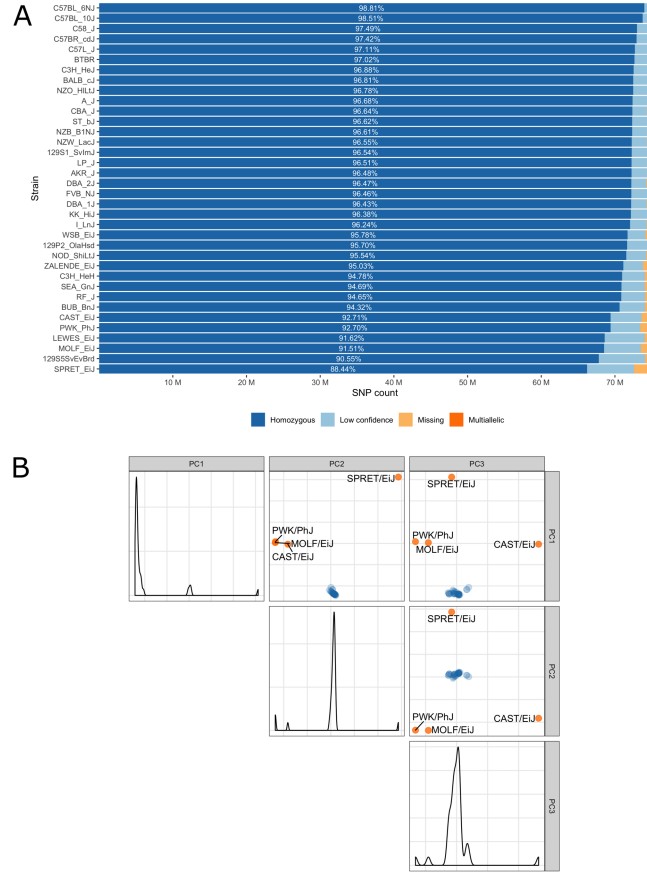

**Figure 3** (A) Inbred mouse strain autosomal SNP characteristics: The number of homozygous, low confidence, missing and multiallelic genotypes for 36 non-reference strains. For each strain, a SNP was checked for group membership in the order low confidence → missing → multiallelic → homozygous → heterozygous and was assigned to the first matching group. Since no SNP made it to the group with heterozygous genotypes it is not shown in the diagram. (B) Principal component analysis shows four outlier inbred strains, CAST/EiJ, PWK/PhJ, SPRET/EiJ and MOLF/EiJ.

## Database

We re-annotated the source VCF file with Ensembl Variant Effect Predictor (VEP) v100 (*McLaren et al., 2016*) using a Docker container image (https://github.com/matmu/vep). For real-time retrieval of variants compatible with phenotypes under various filtering criteria, the variant data was loaded into a MySQL database. The database consists of a single table with columns for chromosomal locus, the reference SNP cluster ID (rsID), variant consequences based on a controlled vocabulary from the sequence ontology (*Eilbeck et al., 2005*), the consequence categorization into variant impacts "HIGH", "MODERATE", 'LOW" or "MODIFIER" according to the Ensembl Variation database (*Hunt et al., 2018*) (see Table S2 for details) and the genotypes (NULL = missing, low confidence, heterozygous or consisting of other alleles than reference or most frequent alternative allele; 0 = homozygous for the reference allele, 1 = homozygous for alternative allele). SNPs with exclusively NULL genotypes were not loaded into the database resulting in

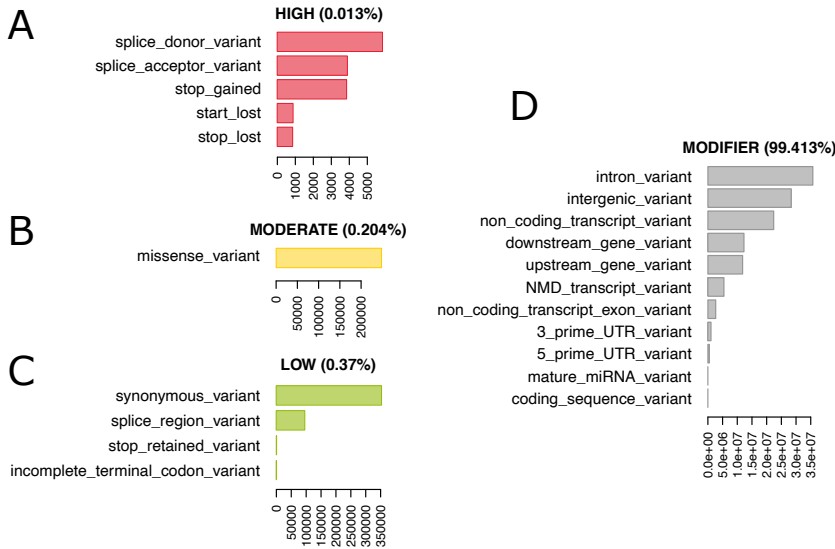

**Figure 4** **74,480,058 variants have been annotated with 120,927,856 consequences. Shown here are the number of variants annotated with a given consequence, stratified by consequence impact.** For description of consequence types see Table S2. Both impact and consequence can be used for variant prioritization in MouseFM. (A) Impact "HIGH"; (B) Impact "MODERATE"; (C) Impact "LOW": (D) Impact "MODIFIER".

74,480,058 autosomal SNVs that were finally added to our database. These have been annotated with overall 120,927,856 consequences, i.e., on average every variant has two annotated consequences. Figure 4 summarizes these consequence annotations stratified by impact; description of consequences and annotation counts are provided in Table S2. Most annotations belong to impact category "MODIFIER" (99.4%). High impact annotations are rare, because they are typically deleterious (0.013%). Annotation with moderate impact consequences comprise only missense, i.e., protein sequence altering variants contributing 0.204%. Low impact consequences are slightly more often annotated, amounting to 0.37%. Ensembl Variant Effect Predictor (VEP) annotation is loaded into the MouseFM database to allow for quick candidate ranking and filtering, which otherwise could not be performed in real-time. Additionally, all candidate variants can be retrieved unfiltered and independent of VEP predictions to allow for custom effect predictions, ranking and filtering.

## Bioconductor R package MouseFM

Our fine-mapping approach was implemented as function `finemap` in the Bioconductor *R* package "MouseFM". Bioconductor is a repository for open software for bioinformatics.

The function `finemap` takes as input two groups of inbred strains and one or more chromosomal regions on the GRCm38 assembly and returns a SNP list for which the homozygous genotypes are discordant between the two groups. Optionally, filters for variant consequence and impacts as well as a threshold for each group to allow for intra-group discordances can be passed. With function `annotate_mouse_genes` the SNP list can further be annotated with overlapping genes. Optionally, flanking regions can be passed.

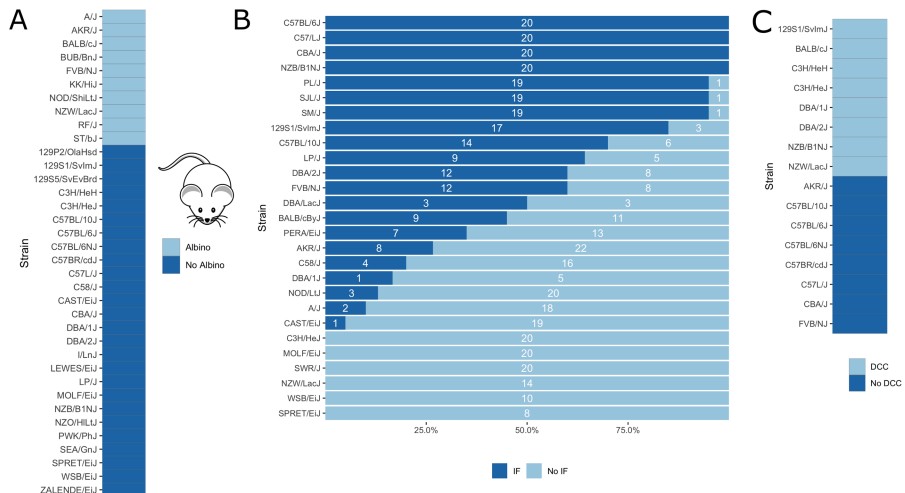

**Figure 5** **Visualization of mouse phenotypic data for which fine-mapping is performed.** (A) Binary in-bred mouse strain phenotype albinism. All or no mice of a strain are albinos; shown here is which strain belongs to which group. (B) Quantitative inbred mouse strain phenotype interfrontal bone (IF). Shown is the number of mice of the respective strain having an interfrontal bone (dark blue, IF) and not having an interfrontal bone (light blue, No IF). (C) Phenotype cardiac dystrophic calcification (DCC). Five inbred strains show the phenotype and five strains lack it.

The `finemap` function queries the genotype data from our backend server while function `annotate_mouse_genes` queries the Ensembl Rest Service (*Yates et al., 2015*). The repository containing the backend of the MouseFM tool, including the scripts of the ETL (Extract, transform, load) process and the webserver, is available at https://github.com/matmu/MouseFM-Backend. Following the repositories' instructions, users may also install the database and server application on a local server.

The workflow and scripts to generate the MouseFM case study results are available at https://github.com/iwohlers/2020_mousefm_finemap.

## RESULTS

In order to characterize fine-mapping results of MouseFM for different numbers of strains and when applying the threshold parameter allowing phenotype-incompatible strains, we used a large gene expression data set. Such a data set contains both (i) genes with clear binary expression phenotype, likely caused by a *cis* variant or haplotype, (ii) cases with no or no binary difference in phenotype.

Further, as a proof of concept, we applied our in silico fine-mapping approach on three additional phenotypes: albinism, interfrontal bone formation and dystrophic cardiac calcification. Phenotypic data is illustrated in Fig. 5.

### Expression quantitative trait loci

MouseFM is particularly useful for detecting variants for which a large, binary effect on a trait can be observed. As such, it is useful for providing candidate variants affecting gene expression, i.e., expression quantitative trait loci (eQTLs). Here, we use two expression

data sets to illustrate this use case as well as to investigate aspects of MouseFM candidate variant lists for a large number of traits with different characterisitics. We use neutorphil and CD4$^+$ T cell expression data from *Mostafavi et al. (2014)* generated in the context of an eQTL study by the Immunological Genome Project. This data is available for 39 inbred mouse strains of which 20 are part of MouseFM. Polymorphonuclear neutrophils (granulocytes) data is available under GEO accession GSE60336, CD4$^+$ T cell data under GSE60337. We downloaded the corresponding normalized expression data from http://rstats.immgen.org/DataPage. Of the strains used here, expression is assessed for two mice each, except for the Black 6J strain of which expression from five mice is available. Neutrophils further have expression for only one FVB mouse.

We read in the expression data and selected all mice from the 20 MouseFM strains ($n = 43$ for CD4$^+$ T cell; $n = 42$ for neutrophils). As *Mostafavi et al. (2014)*, we keep only expressed genes using a cutoff of 120 expression on the intensity scale. This way, we obtain $n = 10,676$ transcripts from 9,136 genes for T cells and $n = 10,137$ transcripts from 8,687 genes for neutrophils, which is comparable to the numbers assessed by *Mostafavi et al. (2014)* using all 39 strains. *Mostafavi et al. (2014)* applied a well-designed dedicated statistical approach to identify and interpret *cis* eQTLs. Briefly, they introduce a metric called TV metric to identify cases of bimodal gene expression and test SNPs within 1 Mb of the transcription start side using a linear regression model. In our experimental setting, we also use a 1 Mb cutoff and aim to detect *cis* eQTLs with MouseFM. For testing, genome-wide 96,779 SNPs were available in the study of *Mostafavi et al. (2014)*. Overall, they identified 1,111 joint T cell and neutrophil eQTLs using $n = 39$ strains. Assessment with MouseFM uses about 74 million SNVs and can be considered somewhat an inverse approach to this previous eQTL study: it is not testing expression differences for a SNP, but it needs as input a separation of strains into two expression groups and identifies all compatible variants, if available. In order to assess characteristics of fine-mapped variants of MouseFM, we use a very crude group separation based on ordering strains using the mouse with minimum expression of a strain and then splitting at the rank of maximum difference between median expression of all mice of strains with smaller rank compared to all mice of strains with larger rank. We run MouseFM for smaller group size from 1 to 10. According to theoretical expectation (cf. Fig. 2), the number of cases in which MouseFM returns candidate variants that are entirely compatible with phenotype decreases with increasing group size, see Fig. 6A for neutrophils and Fig. 6B for CD4$^+$ T cells. At the same time, the proportion of previously detected eQTL transcripts and the number of previously identified eQTL variants increases, because the probability of chance findings decreases. The number of fine-mapped variants varies greatly, often being less than ten but also often more than 100, see Fig. 6C and Fig. 6D, for neutrophils and T cells, respectively. Cases, in which a previously reported eQTL variant was among the fine-mapped variants are comparably few. In these cases, the number of fine-mapped variants tends to be larger than in those cases without a previous eQTL variant among the fine-mapped variants. This effect is likely caused by the much smaller number of variants assessed in the eQTL study –we observe a variant overlap only in cases of large expression-compatible haplotypes. The overall number of fine-mapped variants is rather low, which may be because of the crude
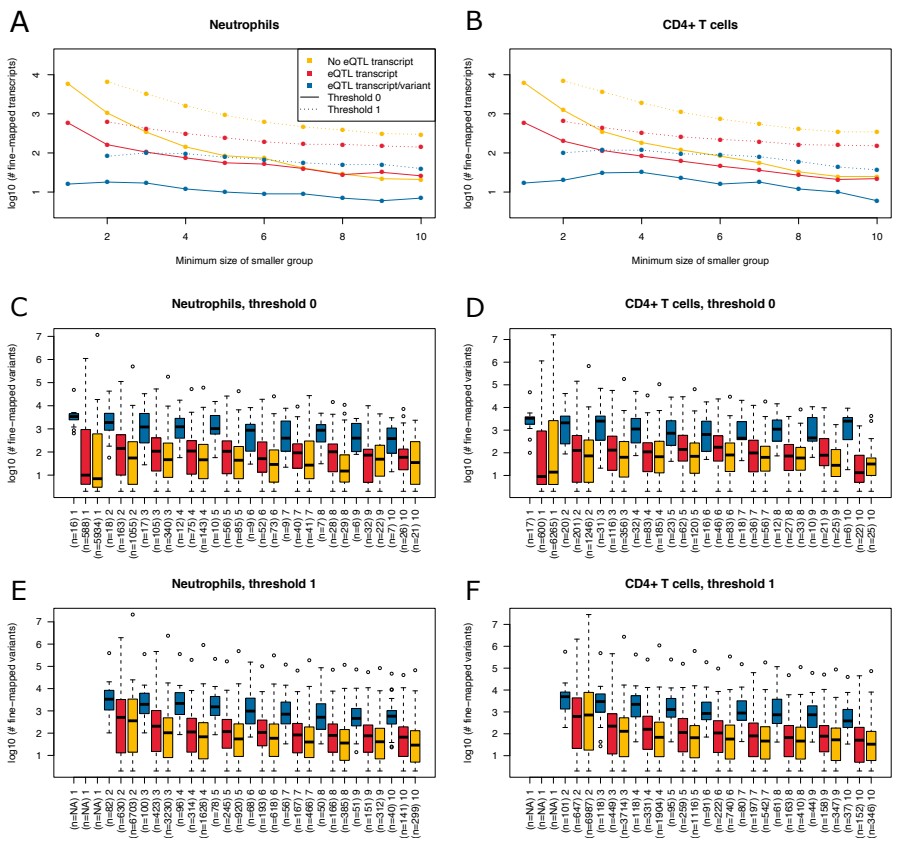

**Figure 6 Summary of fine-mapping results for two expression data sets.** Shown are numbers of fine-mapped transcripts and boxplots of fine-mapped variants for these transcripts. The subset of fine-mapped eQTL transcripts and variants according to *Mostafavi et al. (2014)* is colored blue, the subset of fine-mapped eQTL transcripts without reported eQTL variant according to *Mostafavi et al. (2014)* is colored red, remaining fine-mapped transcripts yellow. (A) The number of successfully fine-mapped transcripts for the neutrophil data set on log 10 scale at different allowed minimum group sizes from 1 to 10. Solid lines denote a threshold of zero incompatible strains, dashed lines denote a threshold of one of incompatible strains (thr1=1 and thr2=1). (B) As A, but for CD4$^+$ T cells. (C) Boxplots of number of fine-mapped variants for the transcripts in A (threshold 0, i.e., solid lines) for different minimum group sizes from 1 to 10. (D) As C, but for CD4$^+$ T cells. (E) Boxplots of number of fine-mapped variants for the transcripts in A (threshold 1, i.e., dashed lines and thr1=1 and thr2=1) for different minimum group sizes from 2 to 10. (F) As E, but for CD4$^+$ T cells.

group definition. We observe that group definition sometimes can be improved, especially if expression is not clearly bimodal. Thus, it is useful to apply MouseFM with a threshold allowing for a given number of incompatible strains. We here allow for one incompatible strain in the first and one incompatible strain in the second group. This increases the number transcripts that could be fine-mapped considerably, especially for large group sizes, see Fig. 6A and Fig. 6B. At the same time, the distributions of number of fine-mapped variants are only marginally affected, see Figs. 6E and 6F. Nearly all high TV scores and/or high effect size and/or low *cis* eQTL *p*-value genes mentioned by *Mostafavi et al. (2014)* can be fine-mapped (71 of 74), illustrating that MouseFM is particularly useful for detecting variants and haplotypes that are compatible with binary, high effect phenotypes.

## Albinism

Albinism is the absence of pigmentation resulting from a lack of melanin and is well-studied in mice (*Beermann, Orlow & Lamoreux, 2004*). It is a monogenic trait caused by a mutation in the *Tyr* gene (*Beermann, Orlow & Lamoreux, 2004*), which encodes for tyrosinase, an enzyme involved in melanin synthesis. The *Tyr* locus has been used before for the validation of in silico fine-mapping approaches (*Cervino et al., 2007*). According to the Jackson Laboratory website (https://www.jax.org), 10 of the 37 inbred mouse strains are albinos with a *Tyr^c* genotype (http://www.informatics.jax.org/allele/MGI:1855976), see Fig. 5A.

Our algorithm resulted in only one genetic locus, which includes the *Tyr* gene; only 245 SNPs have different alleles between the albino and non-albino inbred mouse strains, all located from 7:83,244,464 to 7:95,801,713 (GRCm38). When removing SNPs except those of moderate or high impact, only one variant remains. This variant rs31191169 at position 7:87,493,043, with reference allele C and with alternative allele G in the albino strains is the previously described causal missense SNP in the *Tyr* gene, which results in a cysteine to serine amino acid change at position 103 of the tyrosinase protein.

## Interfrontal bone

Further, we applied our algorithm to the phenotype of interfrontal bone formation, a complex skeletal trait residing between the frontal bones in inbred mice. In some inbred mouse strains, the interfrontal bone is present or absent in all mice, whereas other strains are polymorphic for this phenotype suggesting that phenotypic plasticity is involved. Phenotypic data related to interfrontal bone has recently been generated by *Zimmerman et al. (2019)* for 27 inbred mouse strains (Fig. 5B). They performed QTL mapping and identified four significant loci on chromosomes 4,7,11 and 14, the same loci for interfrontal bone length and interfrontal bone width. For the genotyping, the authors use the mapping and developmental analysis panel (MMDAP; Partners HealthCare Center for Personalized Genetic Medicine, Cambridge, MA, United States), which contains 748 SNPs.

Of the available interfrontal bone data, we only used inbred strains for which all mice show the same phenotype. This corresponds to four strains with interfrontal bone (C57BL/6J, C57L/J, CBA/J, NZB/B1NJ) and five strains without interfrontal bone (C3H/HEJ, MOLF/EiJ, NZW/LacJ, WSB/EiJ, SPRET/EiJ).

In silico fine-mapping resulted in 8,608 SNPs compatible with the observed interfrontal bone phenotype. Of these, 15 showed moderate or high impact on 12 candidate genes, see Table 1. None of the loci identified by us overlaps with the markers of peak LOD score reported by *Zimmerman et al. (2019)* but according to visual inspection, two of their four QTL regions overlap with regions reported by MouseFM, one on chromosome 7 and one on chromosome 11. MouseFM may thus have identified variants underlying those two QTLs. The two other loci reported by *Zimmerman et al. (2019)* may have been missed by MouseFM, because they are driven by strains not used here. Variant rs29393437 is located in the less well described isoform ENSMUST00000131519.1 of *Stac2*, one of two isoforms of this gene. It is is a missense variant, changing arginine (R) to histidine (H) which is at low confidence predicted to be deleterious by SIFT. *Stac2* has been shown to negatively

**Table 1** Moderate and high impact candidate variants and genes for interfrontal bone formation.

| RSID | Position | Gene |
|------|----------|------|
| rs32785405 | 1:36311963 | Arid5a |
| rs27384937 | 2:92330761 | Phf21a |
| rs32757904 | 7:45996764 | Abcc6 |
| rs32761224 | 7:46068710 | Nomo1 |
| rs32763636 | 7:46081416 | Nomo1 |
| rs13472312 | 7:46376829 | Myod1 |
| rs31674298 | 7:46443316 | Sergef |
| rs31226051 | 7:49464827 | Nav2 |
| rs248206089 | 7:49547983 | Nav2 |
| rs45995457 | 9:86586988 | Me1 |
| rs29393437 | 11:98040971 | Stac2 |
| rs29414131 | 11:98042573 | Stac2 |
| rs251305478 | 11:98155926 | Med1 |
| rs27086373 | 11:98204403 | Cdk12 |
| rs27026064 | 11:98918145 | Cdc6 |

regulate formation of osteoclasts, cells that dissect bone tissue (*Jeong et al., 2018*). *Phf21a;* is expressed during ossification of cranial bones in mouse early embryonic stages and has been linked to craniofacial development (*Kim et al., 2012*). Gene *Abcc6* is linked to abnormal snout skin morphology in mouse and abnormality of the mouth, high palate in human according to MGI.

## Dystrophic cardiac calcification

Physiological calcification takes place in bones, however pathologically calcification may affect the cardiovascular system including vessels and the cardiac tissue. Dystrophic cardiac calcification (DCC) is known as calcium phosphate deposits in necrotic myocardial tissue independently from plasma calcium and phosphate imbalances. We previously reported the identification of four DCC loci Dyscal1, Dyscalc2, Dyscalc3, and Dyscalc4 on chromosomes 7, 4, 12 and 14, respectively using QTL analysis and composite interval mapping (*Ivandic et al., 1996*; *Ivandic et al., 2001*). The Dyscalc1 was confirmed as major genetic determinant contributing significantly to DCC (*Aherrahrou et al., 2004*). It spans a 15.2 Mb region on proximal chromosome 7. Finally, chromosome 7 was further refined to a 80 kb region and *Abcc6* was identified as causal gene (*Meng et al., 2007*; *Aherrahrou et al., 2007*). In this study we applied our algorithm to previously reported data on 16 mouse inbred strains which were well-characterized for DCC (*Aherrahrou et al., 2007*). Eight inbred mouse strains were found to be susceptible to DCC (C3H/HeJ, NZW/LacJ, 129S1/SvImJ, C3H/HeH, DBA/1J, DBA/2J, BALB/cJ, NZB/B1NJ) and eight strains were resistant to DCC (CBA/J, FVB/NJ, AKR/J, C57BL/10J, C57BL/6J, C57BL/6NJ, C57BR/cdJ, C57L/J). 2,003 SNPs in 13 genetic loci were fine-mapped and found to match the observed DCC phenotype in the tested 16 DCC strains. Of these, 19 SNPs are moderate or high impact variants affecting protein amino acid sequences of 13 genes localized in two chromosomal regions mainly on chromosome 7 (45.6–46.3 Mb) and 11 (102.4–102.6 Mb), see Table 2. The

**Table 2   Moderate and high impact candidate variants and genes for dystrophic cardiac calcification.**

| RSID | Position | Gene |
|---|---|---|
| rs46174746 | 7:45538428 | *Plekha4* |
| rs49200743 | 7:45634990 | *Rasip1* |
| rs32122777 | 7:45642384 | *Mamstr* |
| rs215144870 | 7:45679109 | *Sec1* |
| rs45768641 | 7:45679410 | *Sec1* |
| rs51645617 | 7:45679423 | *Sec1* |
| rs31997402 | 7:45725284 | *Spaca4* |
| rs50753342 | 7:45794044 | *Lmtk3* |
| rs50693551 | 7:45794821 | *Lmtk3* |
| rs52312062 | 7:45798406 | *Lmtk3* |
| rs49106901 | 7:45798469 | *Emp3* |
| rs47934871 | 7:45918097 | *Emp3* |
| rs32444059 | 7:45942897 | *Ccdc114* |
| rs32753988 | 7:45998774 | *Abcc6* |
| rs32778283 | 7:46219386 | *Ush1c* |
| rs31889971 | 7:46288929 | *Otog* |
| rs50613184 | 11:102456258 | *Itga2b* |
| rs27040377 | 11:102457490 | *Itga2b* |
| rs29383996 | 11:102605308 | *Fzd2* |

SNP rs32753988 is compatible with the observed phenotype manifestations and affects the previously identified causal gene *Abcc6*. This SNP has a SIFT score of 0.22, the lowest score after two SNPs in gene *Sec1* and one variant in gene *Mamstr*, although SIFT predicts all amino acid changes to be tolerated.

## DISCUSSION

With MouseFM, we developed a novel tool for *in silico*-based genetic fine-mapping exploiting the extremely high homozygosity rate of inbred mouse strains for identifying new candidate SNPs and genes. Towards this, by including latest genotype data for 37 inbred mouse strains at a genome-wide scale derived from next generation sequencing, MouseFM uses the most detailed genetic resolution for this approach to date.

Using two large expression data sets, we apply MouseFM to more than 20,000 expression phenotypes of diverse distributions, using different minimum group sizes and also allowing up to one incompatible strain per group. This results in a comprehensive characterization of MouseFM fine-mapped candidate variants. For low group sizes, many phenotype compatible variants can be detected, but these likely include many more false-positives than larger group sies. For larger group sizes, previously identified eQTLs of *Mostafavi et al. (2014)* are much more often successfully fine-mapped than expected by chance, which is in line with theoretical expectation that a given 10/10 group split is rather unlikely to be observed by chance and thus indicates a causal genetic effect. The high number of non-eQTL transcripts that could be fine-mapped also at large group sizes could have several

sources. Firstly, we analyze only 20 strains compared to 39 strains analyzed by *Mostafavi et al. (2014)*, so likely not all of their eQTLs still apply to the smaller set of strains used here. Secondly, previously undetected eQTLs may occur in this smaller set, which could be tested in future work by repeating the *Mostafavi et al. (2014)* analysis for the exact same strains used by MouseFM. Lastly, these may indeed be chance findings unrelated to the expression phenotype, possibly confounded by strain kinship. Manual inspection would help to obtain a clearer picture on a case-by-case basis. Finally, the number of fine-mapped variants varies greatly, so in many cases, additional regulatory information will still be needed to refine the candidate variant list.

By re-analyzing previously published fine-mapping studies for albinism and dystrophic cardiac calBcBcaton, we could show that MouseFM is capable of re-identifying causal SNPs and genes. Re-analyzing a study on interfrontal bone formation (IF) resulted in MouseFM loci that did not overlap the overall markers of peak LOD score reported in the original study, but according to visual inspection, two of the corresponding QTLs. With gene *Stac2* we suggest a new candidate gene possibly affecting interfrontal bone formation.

We selected cases studies particularly to validate that MouseFM can identify experimentally validated variants and genes, such as the *Tyr* variant rs31191169 for albinism and the gene *Abcc6* for dystrophic cardiac calcification. Variant rs31191169 is not a candidate variant and *Abcc6* not a candidate gene, both are experimentally validated to be causally linked to the phenotype. Only for traits that are polygenic, e.g., for DCC (but not for albinism), other candidates returned by MouseFM may be linked to the phenotype, but they do not need to, they are only candidates to follow up on. A different type of case study relates to phenotype interfrontal bone formation, for which causal variants and genes are not known. Still, several candidate genes returned by MouseFM are plausible to affect the phenotype. In summary, additional DCC candidate loci beyond *Abcc6* as well as identified interfrontal bone loci are valid candidate loci. Whether they are in fact affecting the phenotypes needs to be assessed in subsequent QTL and experimental studies.

MouseFM performs most powerful and without limitations for Mendelian traits such as albinism. Secondly, it is most useful as a second-line after QTL mapping. MouseFM is specifically designed to accommodate this fine-mapping setting by allowing to provide start and end of a region to be analyzed. Complex traits and phenotypes with several large effect loci are much more challenging. For these, binarizing the phenotype and performing fine-mapping with MouseFM is not guaranteed to include all causal variants and genes (unlike Mendelian traits). For this reason, we added the option to allow for a user-selected number of outlier strains, which have a genotype discordant with the phenotype. The rationale behind this is identification of genomic regions which are more similar in those strains showing the phenotype compared to strains not showing the phenotype. Lastly, another informative MouseFM setting is the comparison of one phenotype-outlier strain with all other strains, which identifies genetic variants specific to this strain. In summary, MouseFM users need to consider that for polygenic and complex traits, the quality of variant and gene candidates obtained by MouseFM depends on the number and effect size and direction of loci, the genetic diversity of mouse strains and the variability of the phenotype.

A current limitation of MouseFM is that it does only consider single nucleotide variants. Loci containing other types of genetic variation such as insertions, deletions or other, structural variants affecting a phenotype may thus be missed. QTL studies would be able to identify these loci. This could thus be a reason for QTLs without MouseFM support, such as we observe in our case study on interfrontal bone formation. However, this constitutes not a methodological limitation, and other variant types can be added to MouseFM. To date though, genome-wide identification of structural variants is less accurate and less standard compared to small variant identification and thus structural variants are typically not yet systematically analyzed in genetic studies.

We observe that frequently genetic loci identified by MouseFM fine-mapping consist of few or often only a single variant compatible with the phenotype. For example, five of 13 fine-mapped DCC loci comprise a single phenotype-pattern compatible variant and three loci comprise less than 10 variants. This contradicts the expectation that commonly used mice strains differ by chromosomal segments comprising several or many consecutive variants. Commonly used inbred strains display mosaic genomes with sequences from different subspecific origins (*Wade et al., 2002*) and thus one may expect genomic regions with high SNP rate. Fine-mapped loci comprising more phenotype-compatible variants are thus likely more informative for downstream experiments. When allowing no phenotype outlier strain (i.e., thr1=0 and thr2=0), in the case of DCC we identify only six such genetic loci that lend themselves for further experimental fine-mapping (chr7:45,327,763-46,308,368 (811 compatible SNVs); chr7:54,894,131-54,974,260 (32 compatible SNVs); chr9:106,456,180-106,576,076 (170 SNVs); chr11:24,453,006-24,568,761 (40 compatible SNVs); chr11:102,320,611-102,607,848 (46 compatible SNVs); chr16:65,577,755-66,821,071 (890 compatible SNVs)).

## CONCLUSIONS

We show here that in silico fine-mapping can effectively identify genetic loci compatible with the observed phenotypic differences and prioritize genetic variants and genes for further consideration. This allows for subsequent more targeted approaches towards identification of causal variants and genes using literature, data integration, and lab and animal experiments. MouseFM in silico fine-mapping provides phenotype-compatible genotypic differences between representatives of many common laboratory mice strains. These genetic differences can be used to select strains which are genetically diverse at an indicated genetic locus and which are thus providing additional information when performing phenotyping or breeding-based mouse experiments. Thus in silico fine-mapping is a first, very efficient step on the way of unraveling genotype-phenotype relationships.

During the implementation of MouseFM we have paid attention to a very easy handling. To perform a fine-mapping study, our tool only requires binary information (e.g., case versus control) for a phenotype of interest on at least two of the 37 available input strains. Further optional parameters can be set to reduce or expand the search space. MouseFM can also be performed on quantitative traits as we showed for expression data and in the interfrontal bone example.

The general approach underlying MouseFM is straightforward and it has been successfully applied before in a case-wise setting (*Liao et al., 2004*; *Zheng, Dill & Peltz, 2012*; *Hall & Lammert, 2017*; *Mulligan et al., 2019*) and also recently in a high-throughput manner (*Arslan et al., 2020*). Nonetheless, genome-wide variant data of many inbred mouse strains is quite recently available, and this data is large and from raw VCF format difficult to assess systematically for any phenotype of interest. MouseFM is the first tool providing this functionality together with versatile query settings and subsequent variant and gene annotation and filtering options.

In conclusion, MouseFM implements a conceptually simple, but powerful approach for in silico fine-mapping inluding a very comprehensive SNV set of 37 inbred mouse strains. By re-analyzing three fine-mapping studies, we demonstrate that MouseFM is a very useful tool for studying genotype-phenotype relationships in mice. Further, by high-throughput analysis of all genes of two expression datasets, we illustrate that MouseFM is capable of analyzing molecular phenotypes in a versatile and high-throughput manner. This shows the potential of MouseFM to be used for large-scale analyses of diverse phenotypes in future work.

### Funding
This work was supported by the Deutsche Forschungsgemeinschaft (DFG, German Research Foundation) under EXC 22167-390884018, KFO 303 grant number BU 2487/3 and by University of Lübeck, who provided resources at its OMICS compute cluster. The funders had no role in study design, data collection and analysis, decision to publish, or preparation of the manuscript.

### Grant Disclosures
The following grant information was disclosed by the authors:
DFG, German Research Foundation:  EXC 22167-390884018,  KFO 303 r BU 2487/3.
University of Lübeck.

### Competing Interests
The authors declare there are no competing interests.

### Author Contributions
- Matthias Munz, Mohammad Khodaygani and Inken Wohlers conceived and designed the experiments, performed the experiments, analyzed the data, prepared figures and/or tables, authored or reviewed drafts of the paper, and approved the final draft.
- Zouhair Aherrahrou and Hauke Busch conceived and designed the experiments, analyzed the data, authored or reviewed drafts of the paper, and approved the final draft.

### Data Availability
The MouseFM client is available as an R package on GitHub (https://github.com/matmu/MouseFM) and on Bioconductor (https://bioconductor.org/packages/devel/bioc/html/MouseFM.html).

The backend of the MouseFM tool including the scripts of the ETL process and the webserver are available at GitHub: https://github.com/matmu/MouseFM-Backend.

The script used for fine-mapping of expression and of phenotypes albinism, interfrontal bone formation and dystrophic cardiac calcification are also available at GitHub: https://github.com/iwohlers/2020_mousefm_finemap.

## Supplemental Information

Supplemental information for this article can be found online at http://dx.doi.org/10.7717/peerj.11017#supplemental-information.

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
