# Peer review of "In silico candidate variant and gene identification using inbred mouse strains"

_PeerJ, doi:10.7717/peerj.11017_

## Round 0.1 · original submission · Major Revisions

Please pay particular attention to reviewer 3's comments.

Reviewer 1 ·

Basic reporting

This paper presents a tool (MouseFM) that can extract genome-wide all variants that are different between two user specified groups of inbred mouse strains. They downloaded Variant data from Mouse Genomes Project and other public websites and filtered the results.

Major: The bioconductor package doesn’t work and so it could not be tested.

Experimental design

The author needs to provide comparison statistics about MouseFM with others
.

Validity of the findings

They don’t have enough data to support their claim that their tool outperformed others.
The results need to be experimental validated. 
No future work was mentioned

Additional comments

Minor:
There is no description for sub-figure 4C.

Reviewer 2 ·

Basic reporting

The presentation of the material is fine

Experimental design

The design and analysis are straight forward

Validity of the findings

Although there is some replication of previously identified candidate genes, there is no way to know if the novel candidates are any thing more that an association. This makes it difficult to tell if the novel findings are meaningful.

The approach seems to be one that is already used and not necessarily novel.

The conclusions are based on assumptions that predictions made through Vep are accurate and this may not always be the case.

The work would be strengthened if there were some independent means to demonstrate that the new fine-mapped SNP are indeed functional. This is of course a big ask - If the authors pointed out these limitations in the discussion this would certainly be an improvement

Reviewer 3 ·

Basic reporting

The quality of writing is generally good, with a few minor errors, and the article is formatted well. One important omission is that Panel C of figure 4 is not mentioned in the legend.

Experimental design

The aim of the manuscript is well described, and the research gap is pointed out, i.e. that genotypes from large sequencing datasets and large phenotype datasets have not been well integrated.
I have concerns about the methods, and how they were shown to be significant, which I have described in my general comments.

Validity of the findings

The results presented are valid, but I have concerns about how generalizable they are, as outlined in my general comments. The underlying data and code is available, and the R code is easy to use.

Additional comments

Dear Authors,
The manuscript is interesting, and provides a good interface for looking for variants between inbred strains. As the authors point out, there is now good phenotype and genotype data for many of these inbred strains, and these two resources should be taken advantage of. The manuscript is well written, with the authors points being clearly made.

That being said, I have a few concerns that I feel need to be addressed.

It is clear to me that this method works for mendelian traits, such as albinism used by the authors. It is much less clear how well this will work for quantitative traits, especially those that have several causal variants of similar effect size. If it is used as a second-line after QTL mapping (i.e. just looking at between strain differences within the QTL region) then I see how it would work, but as presented, it appears to be looking genome-wide from the start and clustering the strains into two groups, which would simply not work for cases with several large effect loci.

I think rather than just looking at three examples, a more convincing argument would be made if a whole dataset (e.g. all of the JAX mouse phenome data with > X sequenced strains, or a large gene expression dataset) was used.

Linked to this is a concern about power. Given the small number of strains (e.g. for interfrontal bone formation 9 strains) and the large number of variants (up to 74,873,854) how many variants would we expect to see segregating between the two groups by chance? A simple way of testing this would be to look at the number of variants called by your method in each combination of four and five strains, and how many of these meet the criteria that you used to prioritize genes for presence of the interfrontal bone. My worry is that you will get nearly as many candidate variants using a random split of strains as you got by splitting strains by phenotype.

These concerns are somewhat demonstrated in the authors’ Dystrophic cardiac calcification use case. In the previous QTL mapping, four loci were identified, whereas using the current ‘fine mapping’ method, 13 loci were identified. It is unclear to me how the authors prove that the additional 9 loci are not simply false positives.

Another issue is that only SNPs are used. Although these are easier to find in commonly used short-read sequencing data, it leaves all potentially larger variants (e.g. insertions and deletions), that could also alter the phenotypes, unaccounted for. In QTL mapping this is not such an issue, since the markers/SNPs will tag nearby large variants, but in this case, where variants (and therefore genes) are being prioritized based on their predicted effects then this is not the case.

Linked to this, variants are being prioritized based on their protein coding effects (e.g. ‘high’ impact). GWAS data in humans shows the importance of gene expression regulating variants. These variants would be missed by the current method. As a demonstration of this, one wonders how this method would perform in providing candidate variants for gene expression as a phenotype.

Finally, there does not appear to be any consideration of potential kinship between inbred strains. Given the relatedness among many inbred strains, it is possible that situations occur where strains share many variants simply due to this kinship. This becomes increasingly problematic as fewer strains are used for phenotyping. The inverse of this is that if closely related strains show different phenotypes, then there will be fewer variants between the two (and examples of this have been used for reduced complexity crosses). This effect of kinship should be mentioned.


Smaller, minor suggestions are below:

In the abstract, ‘In addition, lately, also’ does not read well, ‘In addition, recently’ would work better.

Line 35: ‘favourable animal model’ should be plural, ‘favourable animal models’

Lines 38-40. It would perhaps be better to refer to this strain as Black 6J. C57BL/6J, C57BL/6N, C57BL/6NJ and C57BL/6NCrl are all sometimes called Black 6, and indeed, you have C57BL/6N among your inbred strains.

Lines 47-50: This idea of in silico fine mapping has certainly been used since then, even if not using that exact phrase. Examples include Hall and Lammert, 2017 (10.1007/978-1-4939-6427-7_21) and Mulligan et al., 2019 (10.3389/fgene.2019.00188).

Line 109: End of parentheses missed

Lines 137: The sentence should start with ‘The’, i.e. ‘The function finemap takes’.

Line 149: “albinisim” should be “albinism”

Figure 4: Panel C is not mentioned in the legend.

Line 166: Should read ‘we applied our algorithm to the phenotype’

---

## Round 0.2 · Minor Revisions

I apologize for the delay, There are some minor issues raised by reviewer 3. Could you please address them quickly and I promise I will give a quick turn around once I received your revision. Thank you.

Reviewer 3 ·

Basic reporting

Well written throughout, with no major errors.

Experimental design

The aim of the manuscript is well described, and the research gap is pointed out, i.e. that genotypes from large sequencing datasets and large phenotype datasets have not been well integrated.
The authors have addressed or at least acknowledged previous concerns about significance and power, so that they are now clear to the reader.

Validity of the findings

Again, the authors have addressed my concerns abut the validity of the findings, and pointed out the caveats in a way that is clear to the reader.
There is still some question about the novelty of the biological findings (as they are not validated), but I consider this to be secondary to the main point of the paper: the authors have produced an easily used code for looking for variants between strains that they can expand in the future, and shown that it gives useful outputs.

Additional comments

Dear Authors,
I feel that you have addressed the review comments well, in that you have made clear to the reader what the short-comings are, and how to minimize false discoveries.


Smaller, minor suggestions are below:

Lines 62-63: The preprint Ashbrook et al was recently published, PMID: 33472028 https://t.co/gDeKS50ANt?amp=1

Lines 69-70: I think it would read better if the sentence was “Several of the aforementioned resources allow the user to interactively query genotypes for 70 user-selected inbred mouse strains for input genes or genetic regions”.

Lines 70-72: The Variant browser in GeneNetwork has some of this functionality (http://www.genenetwork.org/snp_browser and http://gn1.genenetwork.org/snpbrowser.html), however it does not seem to be programmatic accessible, limiting its usefulness compared to MouseFM.

Line 181: Should read “we used” rather than “we use”.

Expression quantitative trait loci section: It would be interesting to know how many of the novel (i.e. not found in Mostafavi) variants are possibly cis (e.g. with ~5-10Mb of the gene)? I would expect many of the these large effect variants to be in cis. This might be another quality metric, or another way to determine the best candidate gene for expression of that gene.

Lines 263-272: Looking at the Zimmerman et al 2019 paper, it seems that they don't give confidence intervals (e.g. 1.5 LOD drop), but rather just gave the peak position. Eyeballing their QTL maps (without having the data to reanalyze it), it seems like your Chr7, and certainly your Chr11 SNPs overlap with the QTL detected in Zimmerman et al.
To me, it would be a plausible suggestion that you have identified variants underlying those two QTL. It is also possible that the two QTL that you do not see in this study are driven by the other strains that you did not use.

Lines 289-290: It seems odd that you detect the same QTL region on chr7 for Dystrophic cardiac calcification and Interfrontal bone. Is this due to chance? Biological overlap (e.g. something in the calcification/bone pathway)? Or similar strains, and therefore regions that are divergent between the groups appear often?

Line 301: “a setting” is not needed.

Line 310-311: If the data for the eQTL mapping was available, you could redo it, using only your subset of strains, to test this hypothesis.

Line 379-380: the expression datasets are also an example of quantitative traits.

---

## Round 0.3 · accepted · Accept

Thank you for your quick response.

---

## Author Rebuttal · Round 0.3

Thanks for the suggestions from reviewer 3, which we all implemented in the submitted minor revision as detailed in the following.

## Reviewer 3

**Basic reporting**

Well written throughout, with no major errors.

Thank you.

**Experimental design**

The aim of the manuscript is well described, and the research gap is pointed out, i.e. that genotypes from large sequencing datasets and large phenotype datasets have not been well integrated.
The authors have addressed or at least acknowledged previous concerns about significance and power, so that they are now clear to the reader.

Thank you.

**Validity of the findings**

Again, the authors have addressed my concerns abut the validity of the findings, and pointed out the caveats in a way that is clear to the reader.
There is still some question about the novelty of the biological findings (as they are not validated), but I consider this to be secondary to the main point of the paper: the authors have produced an easily used code for looking for variants between strains that they can expand in the future, and shown that it gives useful outputs.

Thank you.

**Comments for the author**

Dear Authors,
I feel that you have addressed the review comments well, in that you have made clear to the reader what the short-comings are, and how to minimize false discoveries.

Thank you.

Smaller, minor suggestions are below:

Lines 62-63: The preprint Ashbrook et al was recently published, PMID: 33472028
https://t.co/gDeKS50ANt?amp=1

We updated the reference.

Lines 69-70: I think it would read better if the sentence was "Several of the aforementioned resources allow the user to interactively query genotypes for 70 user-selected inbred mouse strains for input genes or genetic regions".

We updated the sentence as suggested.

Lines 70-72: The Variant browser in GeneNetwork has some of this functionality (http://www.genenetwork.org/snp_browser and http://gn1.genenetwork.org/snpbrowser.html), however it does not seem to be programmatic accessible, limiting its usefulness compared to MouseFM.

We added a corresponding sentence as follows:
*Moreover, the variant browser in GeneNetwork allows also comparison of genotypes between strains, however, data can only be extracted gene- or region-wise and is not accessible programmatically.*

Line 181: Should read "we used" rather than "we use".

Corrected.

Expression quantitative trait loci section: It would be interesting to know how many of the novel (i.e. not found in Mostafavi) variants are possibly cis (e.g. with ~5-10Mb of the gene)? I would expect many of the these large effect variants to be in cis. This might be another quality metric, or another way to determine the best candidate gene for expression of that gene.

We agree. In the current experimental setting though, we only aim to identify cis eQTLs, and thus assessed the same region as in Mostafavi *et al*., i.e. variants within 1 MB of the transcription start site. We made this clearer by adding the sentence:
*In our experimental setting, we also use a 1 MB cutoff and aim to detect cis eQTLs with MouseFM.*

Lines 263-272: Looking at the Zimmerman et al 2019 paper, it seems that they don't give confidence intervals (e.g. 1.5 LOD drop), but rather just gave the peak position. Eyeballing their QTL maps (without having the data to reanalyze it), it seems like your Chr7, and certainly your Chr11 SNPs overlap with the QTL detected in Zimmerman et al.
To me, it would be a plausible suggestion that you have identified variants underlying those two QTL. It is also possible that the two QTL that you do not see in this study are driven by the other strains that you did not use.

We agree, and thanks a lot for investigating this closer. So far, we considered the signals not to overlap, because their peak position is not contained within the MouseFM reported loci.

It is true though, that according to Fig. 4 of Zimmerman *et al*. the QTL regions are much larger and likely include the MouseFM loci on chromosome 7 and 11. We updated the sentence accordingly as suggested:

*None of the loci identified by us overlaps with the markers of peak LOD score reported by Zimmerman et al., but according to visual inspection, two of their four QTL regions overlap with regions reported by MouseFM, one on chromosome 7 and one on chromosome 11. MouseFM may thus have identified variants underlying those two QTLs. The two other loci reported by Zimmerman et al. may have been missed by MouseFM, because they are driven by strains not used here.*

Accordingly, we updated lines 317-319, which now read:

*Re-analyzing a study on interfrontal bone formation (IF) resulted in MouseFM loci that did not overlap the markers of peak LOD score reported in the original study, but according to visual inspection, two of the corresponding QTLs.*
* * *
Lines 289-290: It seems odd that you detect the same QTL region on chr7 for Dystrophic cardiac calcification and Interfrontal bone. Is this due to chance? Biological overlap (e.g. something in the calcification/bone pathway)? Or similar strains, and therefore regions that are divergent between the groups appear often?
* * *
Yes, we were also wondering whether this is driven by a shared biological mechanism, strain similarity, or by chance. Shared biology is plausible, as well as strain group similarity. However, additional lab and in silico experiments are needed to investigate this closer.
* * *
Line 301: "a setting" is not needed.
* * *
We removed it.
* * *
Line 310-311: If the data for the eQTL mapping was available, you could redo it, using only your subset of strains, to test this hypothesis.
* * *
This is true, we updated the sentence accordingly:

*Secondly, previously undetected eQTLs may occur in this smaller set, which could be tested in future work by repeating the Mostafavi et al. analysis for the exact same strains used by MouseFM.*
* * *
Line 379-380: the expression datasets are also an example of quantitative traits.
* * *
Yes, we thus updated the sentence to:

*MouseFM can also be performed on quantitative traits as we showed for expression data and in the interfrontal bone example.*